

# Interaction and mechanisms of depression and cardiovascular disease: a mini-review

Ling Huang[1,*], Lei Zhang[2,*], Cuihua Liu[1], Qiang Xu[2] and Kuncheng Qiu[1]

[1] Third-grade Pharmacological Laboratory on Traditional Chinese Medicine, State Administration of Traditional Chinese Medicine, College of Medicine and Health Sciences, China Three Gorges University, YiChang City, China
[2] Department of Cardiology, Yichang City Hospital of Traditional Chinese Medicine, YiChang City, China
[*] These authors contributed equally to this work.

## ABSTRACT

Cardiovascular disease (CVD) is the world's leading cause of death and frequently comorbid with depression. Numerous clinical studies reveal a bidirectional interaction between CVD and depression, mutually exacerbating the pathological progression of both conditions. This comorbidity has emerged as a significant global health challenge, yet targeted therapeutic drugs and strategies remain underdeveloped. This article summarizes the relationship between depression and various CVD types, including coronary heart disease, atherosclerosis, myocardial infarction, and heart failure. We further analyze the underlying mechanisms through autonomic nervous system dysfunction, hypothalamic-pituitary-adrenal axis dysregulation, inflammatory responses, endothelial dysfunction, and platelet activation pathways. By integrating these insights, this review aims to provide references for the development of novel therapeutics for CVD combined with depression.

## INTRODUCTION

Recent epidemiological surveillance data from the World Health Organization (WHO) indicates that cardiovascular diseases (CVDs) are now the top cause of death worldwide, responsible for about 32% of all global death (*WHO, 2024*). CVDs, such as coronary heart disease (CHD), atherosclerosis (AS), acute coronary syndrome (ACS), myocardial infarction (MI), and heart failure (HF), not only severely compromise patients' quality of life but also pose life-threatening risks (*Tong et al., 2024*).

Over recent years, many studies have revealed a strong association between CVD and depression. Depression not only significantly exacerbates the clinical progression but also serves as an independent predictor of major adverse cardiovascular events (*Correll et al., 2017*). Notably, patients who suffered CVD combined with depression exhibit 10% to 40% higher incidence and mortality rates compared to those with either condition alone, and this risk increases with the severity of the condition (*Pinter et al., 2019*). The pathogenesis of CVD-depression comorbidity is widely recognized as complex and multifaceted,

Corresponding authors
Qiang Xu, 9463488@qq.com
Kuncheng Qiu, qiukun108@126.com

involving inflammatory responses, dysfunction of the hypothalamic-pituitary-adrenal (HPA) axis, the autonomic nervous system, and other mechanisms (*Xu et al., 2024*). However, the bidirectional interplay between CVD and depression complicates therapeutic outcomes, as drugs targeting either condition alone often fail to achieve satisfactory efficacy. Consequently, there remains a critical lack of effective treatment strategies for CVD combined with depression in clinical practice.

We think that elucidating the relationship between CVD and depression and deciphering their mechanisms of mutual influence are essential for developing targeted therapies. Therefore, this review synthesizes current evidence regarding the bidirectional interactions and comorbid mechanisms between CVD and depression, providing cardiologists, psychiatrists/psychologists, basic medical researchers, and clinicians focused on comorbidity management with updated insights to develop targeted therapies and optimize clinical management.

## SURVEY METHODOLOGY

Given that this article aims to provide a comprehensive overview of existing evidence and underlying mechanisms regarding the interplay between CVD and depression—without conducting rigorous quantitative analysis—we adopted a narrative review methodology to synthesize the literature. The literature search was conducted across the PubMed, Web of Science, and Chinese National Knowledge Infrastructure (CNKI) databases. The search encompassed articles published within the past five years and earlier relevant publications. Keywords included: depression and cardiovascular disease, mechanisms of cardiovascular disease, mechanisms of depression, depression and coronary heart disease, depression and MI, depression and AS, depression and HF, *etc.* As the research advanced, the search strategy was refined using additional keywords such as: inflammatory factors and cardiovascular disease, inflammatory factors and depression, oxidative stress and cardiovascular disease, depression and the autonomic nervous system, depression and the HPA axis, drugs with antidepressant effects, *etc.* Following removal of duplicates and irrelevant articles, 91 articles were ultimately included in this review.

## DEPRESSION AND CVD

### Depression and CHD

CHD is characterized by atherosclerotic changes in the coronary arteries, leading to vascular lumen obstruction or narrowing, which subsequently causes myocardial ischemia, hypoxia, and even necrosis (*Abdul Manan et al., 2024*). As of 2023, the global population of CHD patients reached approximately 11.39 million, with CHD remaining the leading cause of death among patients with CVD. Concurrently, depression ranks as the second leading cause of death worldwide, following CHD. Depression is also an independent risk factor for adverse cardiovascular events (*Yusuf et al., 2020*). The close association between the two conditions has garnered widespread attention, as illustrated in Fig. 1.

A bidirectional interaction exists between depression and CHD. On the one hand, depression significantly accelerates the onset and advancement of CHD. Studies indicate

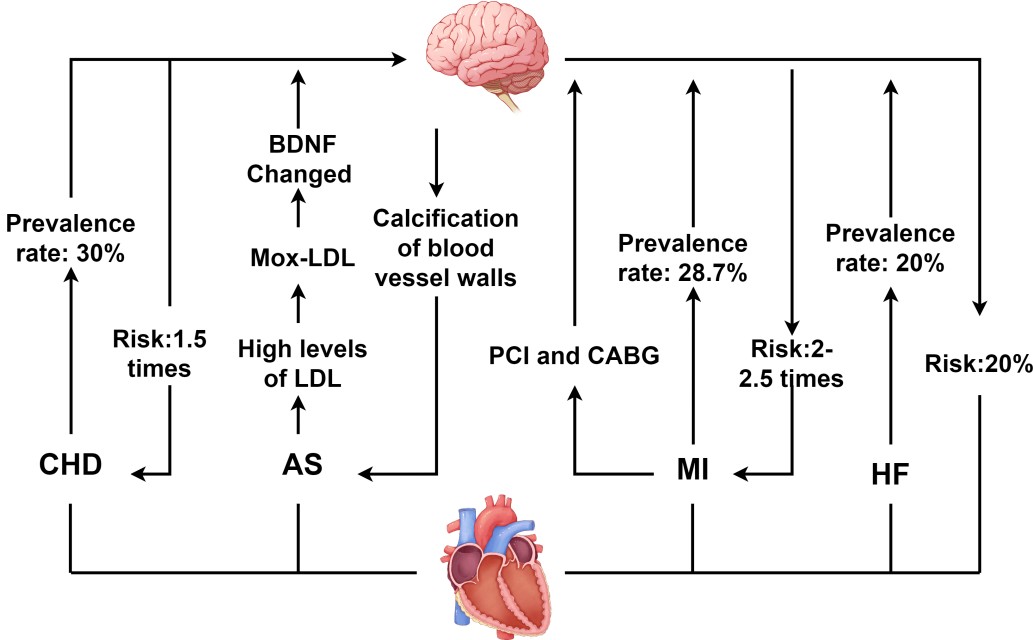

**Figure 1  Interaction between depression and CVD.** CHD, coronary heart disease; AS, atherosclerosis; MI, myocardial infarction; HF, heart failure; PCI, percutaneous coronary intervention; CABG, coronary artery bypass grafting; LDL, low density lipoprotein; BDNF, Brain derived neurotrophic factor. Created with Figdraw 2.0: https://www.figdraw.com.

that depression elevates CHD risk by 1.5-fold, while severe depression increases the risk by 2.7-fold (*Jha et al., 2019*). Additionally, depression increases the incidence of both stable and unstable angina as well as MI in CHD patients (*Abdul Manan et al., 2024*; *Daskalopoulou et al., 2016*). On the other hand, CHD itself and its treatments may induce or exacerbate depressive symptoms. Epidemiological data reveal that depression prevalence is significantly higher among CHD patients, with approximately 30% exhibiting clinically significant depressive symptoms, and 15% to 20% meeting diagnostic criteria for major depressive disorder (MDD) (*Daskalopoulou et al., 2016*). This complex bidirectional relationship between CHD and depression poses significant challenges for clinical treatment of patients with comorbid CVD and depression.

## Depression and AS

AS involves arterial narrowing or occlusion, primarily caused by the accumulation of plaques containing lipids, low-density lipoprotein (LDL), and cholesterol (*Libby et al., 2019*). It is a major contributor to both cardiovascular and cerebrovascular diseases (*Libby, 2021*). When AS affects the coronary arteries, plaque enlargement and instability can obstruct coronary vessels, leading to myocardial ischemia and subsequent CHD. Clinical evidence indicates that the atherogenic index of plasma (AIP) positively correlates with depression (*Kong & Zou, 2024*), and patients with depression exhibit higher susceptibility to AS development (*Seldenrijk et al., 2010*). Concurrently, in AS patients, elevated LDL levels undergo myeloperoxidase-mediated oxidation to form oxidized LDL (Mox-LDL).

Mox-LDL modulates brain tissue plasminogen activator (tPA) activity *via* the lectin-like oxidized LDL receptor-1 (LOX-1) signaling pathway and regulates neuroserpin serine protease activity. This process disrupts brain-derived neurotrophic factor (BDNF) processing, impairing neuroplasticity and promoting depression pathogenesis (*Daher, 2024*; *Frangie & Daher, 2022*).

Furthermore, when unstable plaques in the coronary arteries rupture or erode, they can cause thrombosis, embolism, or vasospasm, leading to reduced coronary blood flow and triggering ACS. Notably, individuals with depression face a threefold higher risk of ACS than non-depressed populations (*De Giorgi et al., 2021*). Among ACS patients, the prevalence of depression reaches approximately 66.3% (*Zou et al., 2023*). Depression not only accelerates coronary thrombus formation but also increases the relative risk of venous thromboembolism, adversely impacting prognosis of ACS (*Lin et al., 2019*).

## Depression and MI

MI results from coronary artery occlusion caused by conditions such as aortic dissection, ACS, or atherosclerotic plaque rupture. It manifests with severe chest pain and may be accompanied by HF symptoms such as dyspnea and cough. The severity of MI varies depending on the location and extent of coronary involvement (*Han et al., 2023*; *Yıldırım & Kocatepe, 2023*). Studies have shown that approximately 28.7% of MI patients suffer from depression, with a prevalence of major depressive disorder (MDD) of about 10% (*Feng et al., 2019*). Depression elevates the risk of cardiovascular complications in MI patients by 2- to 2.5-fold and increases mortality risk by 2.7-fold (*Feng et al., 2019*; *Henderson et al., 2024*). When MI occurs, restoring myocardial blood supply through percutaneous coronary intervention (PCI) or coronary artery bypass grafting (CABG) represents an effective therapeutic approach (*Emamzadehashemi et al., 2024*; *Meng et al., 2020*; *Savic et al., 2023*). However, these interventions may themselves contribute to depression onset. Studies indicate that the incidence of post-PCI depression is as high as 16.6%, while among CABG patients, 26% experience mild depression and 12% suffer from moderate to severe depression (*Gutlapalli et al., 2022*; *Jha et al., 2019*). Additionally, PCI may induce myocardial ischemia-reperfusion injury, exacerbating vascular inflammatory responses. This process adversely affects the nervous system, thereby promoting depression pathogenesis.

## Depression and HF

HF results from compromised cardiac pumping function, with typical causes including MI, cardiomyopathy, and myocarditis. HF not only imposes a heavy burden on the healthcare system but also has a high mortality rate, posing a serious threat to patients' lives (*Savarese et al., 2023*; *Wang et al., 2021*). Depression and HF also exhibit a bidirectional relationship. Depression increases HF risk by approximately 20% and accelerates disease progression in affected patients (*Celano et al., 2018*). It is also an independent predictor of poor prognosis in established HF (*Silverman, Herzog & Silverman, 2019*).

Studies indicate that the probability of depression in HF patients is 20%, a rate at least threefold higher than in the general population. Moreover, depressed HF patients

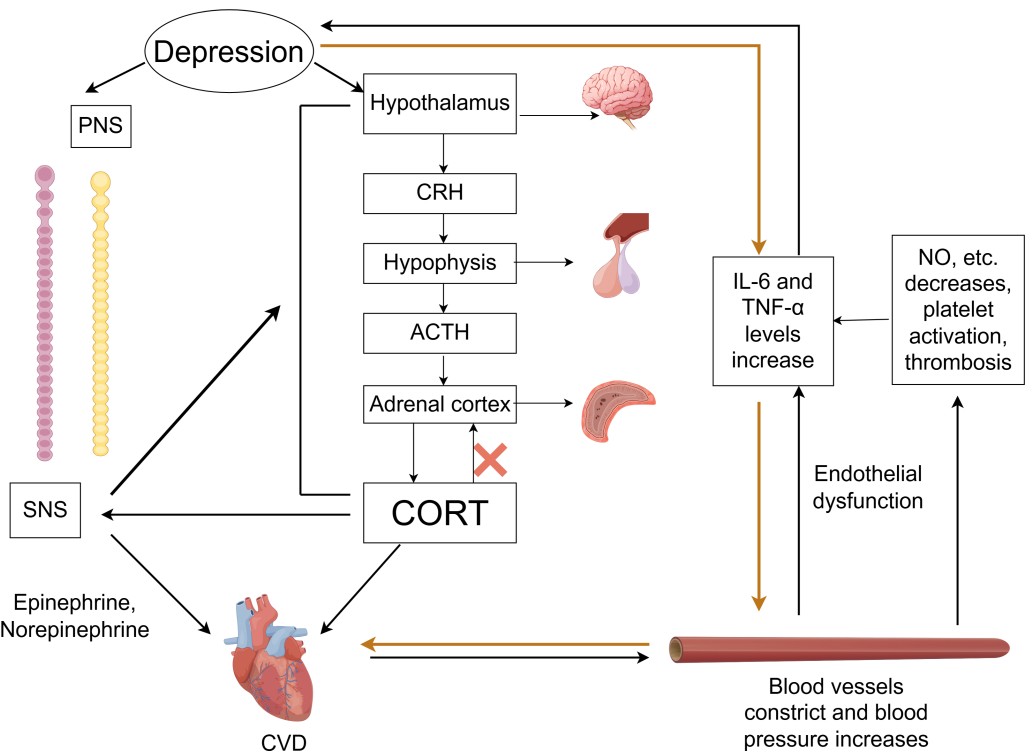

**Figure 2  Comorbidity mechanisms between depression and CVD mediated through the ANS, HPA axis, inflammatory response, endothelial dysfunction and platelet activation.** CVD, cardiovascular diseases; SNS, the sympathetic nervous system; PNS, the parasympathetic nervous system; CRH, corticotropin-releasing hormone; ACTH, adrenocorticotropic hormone; CORT, cortisol. Created with Figdraw 2.0: https://www.figdraw.com.

face a mortality risk as high as 60% (*Bahr et al., 2024*; *Sbolli et al., 2020*). Although the impact of depression on HF is well recognized, the impact of HF on depression remains incompletely understood. The prevailing view is that HF reduces cerebral blood flow, triggers neuroinflammatory responses, and subsequently affects brain functional areas such as the hippocampus and amygdala, ultimately contributing to depression (*Díaz et al., 2020*; *Toledo et al., 2019*).

## COMORBID MECHANISMS OF CVD AND DEPRESSION

### Dysfunction of autonomic nervous system
In the comorbidity mechanisms between CVD and depression, autonomic nervous system dysfunction plays a crucial role, as shown in Fig. 2. The autonomic nervous system is primarily composed of two parts: the sympathetic nervous system (SNS) and the parasympathetic nervous system (PNS). Under physiological conditions, the balance between the SNS and PNS helps maintain the stability of the body's internal environment. Depression may cause autonomic dysfunction and hormonal imbalances, which in turn can have adverse effects on CVD (*Bremner et al., 2018*).
Imaging studies have identified changes in the amygdala, hippocampus, medial prefrontal cortex, and anterior cingulate cortex in the brains of those with depression (*Vaccarino et al., 2020*). These brain regions are responsible for asymmetric sympathetic input to the heart. The SNS initiates a signaling cascade by binding norepinephrine (released from neurons) and epinephrine (secreted by chromaffin cells of the adrenal medulla) to β-adrenergic receptors on the myocardial cell membrane. Subsequent cAMP generation and protein kinase A activation regulate intracellular calcium flux in cardiomyocytes, altering cardiac contractility and rate—thereby accelerating CVD progression (*Del Rivero Morfin, Marx & Ben-Johny, 2023*). Taking HF as an example, excessive activation of the SNS is a key feature, which further intensifies as HF progresses. This can lead to arrhythmias and sudden cardiac death (*Li, 2022b*; *Triposkiadis et al., 2024*). The progression of HF, in turn, forms a feedback loop with the SNS system. During this process, neurohormonal changes may disrupt neurotransmitter systems, alter brain function, and affect emotional regulation, thereby promoting the development of depression in return (*Rashid et al., 2023*).

Currently, heart rate variability (HRV) is commonly used in clinical to indirectly evaluate the relationship between CVD and the autonomic nervous system. HRV reflects variations in beat-to-beat intervals, serving as a key indicator of sympathetic and parasympathetic nerve activity and balance (*Hong et al., 2022*; *Khawaja et al., 2009*; *Walter et al., 2019*). Patients with good cardiac function typically exhibit higher HRV. Compared to non-depressed CAD patients, those with depression demonstrate significantly lower HRV. Notably, HRV is highly susceptible to influences from age, gender, respiratory rate, body position, medications (particularly antidepressants), and lifestyle factors. The current lack of standardized measurement protocols directly contributes to substantial heterogeneity across studies. Therefore, the relationship between HRV and depression combined with CVD still need further research.

## Dysfunction of the HPA axis

As a vital part of the neuroendocrine system, the HPA axis plays a critical role in the mutual influence between CVD and depression. Within the HPA axis, neurons in the paraventricular nucleus (PVN) of the hypothalamus synthesize and secrete corticotropin-releasing hormone (CRH). This hormone is transported to the anterior pituitary *via* the hypothalamic-pituitary portal system, stimulating corticotrophs to release adrenocorticotropic hormone (ACTH). ACTH subsequently acts on the adrenal cortex to stimulate glucocorticoid synthesis and secretion—primarily cortisol (CORT).

Activation of the HPA axis in depression triggers hypothalamus secretion of CRH, which stimulates pituitary release of ACTH, resulting in higher CORT levels (*Menke et al., 2021*). Under physiological conditions, when CORT levels reach saturation, they activate the glucocorticoid receptors (GRs), stimulating the hippocampus to send negative feedback signals to regulate the HPA axis and reduce CORT secretion. However, chronic or acute elevation of CORT leads to excessive GR activation, resulting in hippocampal damage and impairment of the HPA axis negative feedback regulation—culminating in HPA axis hyperfunction (*He, Zhang & Chen, 2016*). This dysfunction establishes a vicious cycle: sustained CORT elevation exacerbates hippocampal injury, disrupts neurotransmitter

homeostasis, and impairs neural circuitry function, thereby intensifying depressive symptoms (*Cernackova et al., 2020*; *Rashid et al., 2023*). Concurrently, elevated CORT not only worsens depression but also induces vasoconstriction, increases blood pressure, and heightens cardiovascular sensitivity to catecholamines. These effects promote excessive release of proinflammatory cytokines (*e.g.*, TNF-α, IL-1β, IL-6) (*Fioranelli et al., 2018*). The resultant inflammatory response accelerates AS, hypertension, myocardial ischemia-reperfusion injury, and other CVDs (*Khawaja et al., 2009*; *Liu et al., 2023*). Notably, it has been reported that inhibiting the activity of CRH neurons in PVN can ameliorate CVD pathogenesis (*Zhou et al., 2024*).

Additionally, the HPA axis exhibits close interplay with the SNS. The overactivation of the SNS can stimulate the HPA axis, leading to increased CORT release, while elevated CORT levels potentiate sympathetic activity. This bidirectional interaction can amplify neuroendocrine dysregulation, exacerbate symptoms of anxiety and depression, and increase the risk of CVD. However, evidence quantifying changes in CORT levels in relation to the risk of specific cardiovascular events remains limited. Future research requires more precise biomarkers to elucidate the dynamic changes in HPA axis activity during comorbidity progression.

## Inflammatory response

Inflammation represents a defensive response of the body to external or internal damage, characterized by inflammatory cell infiltration into damaged tissues, clearance of cellular debris, activation of immune system, and the release of chemokines and cytokines, which promote fibroblast activation and tissue repair (*Varghese et al., 2024*).

Recent studies have shown that inflammation plays a significant role in CVDs—including CHD, AS, MI, and HF—while also revealing its bidirectional association with the occurrence and progression of depression (*Lin et al., 2023*; *Mason & Libby, 2015*).

Within the central nervous system, depression is often accompanied by the activation of central inflammatory responses (*Yu et al., 2021*). Inflammatory cytokines can disrupt the integrity of neurotransmitter signaling in the cerebral cortex and hippocampus, leading to imbalances in neurotransmitters such as GABA, glutamate and 5-HT, which contribute to psychological disorders and further exacerbate the severity of depression (*Troubat et al., 2021*; *Wu et al., 2020*). The interaction between inflammation and psychological disorders creates a vicious cycle. As previously noted, depression dysregulates the HPA axis and also triggers inflammatory responses, thereby amplifying CVD risk.

For patients suffering from CVD combined with depression, depression often leads to an inflammatory response, manifested by increased levels of inflammatory factors, such as IL-6, TNF-α, hypersensitive C-reactive protein, and ICAM-1, *etc*. The high levels of inflammatory factors not only mediate nonspecific inflammatory responses and alter immune reactivity but also have significant impacts on CVD (*Attiq et al., 2024*; *Müller, 2019*; *Wang et al., 2017*). To put it more concretely, elevated IL-6 levels are significantly associated with increased risks of MI and cardiovascular mortality (*Patas et al., 2014*; *Ridker & Rane, 2021*). TNF-α, IL-1β, and C-reactive protein promote the atherosclerotic lesion progression by facilitating foam cells formation, accelerating intraplaque hemorrhage, and inducing

the secretion of matrix-degrading enzymes (*Kalkman, 2020*; *Yang et al., 2021*). HighIL-1βlevels can induce both neuroinflammation and peripheral inflammation, potentially driving pathological changes—including cardiac remodeling, left ventricular dilation, and fibrosis—through long-term chronic inflammatory response, thereby compromising long-term cardiac function (*Ardinal, Wiyono & Estiko, 2024*; *Kouba et al., 2024*). Additionally, these cytokines can directly affect myocardial contractility and induce cardiomyocyte apoptosis (*Xu et al., 2024*).

Given the critical role of inflammation in CVD with depression, interventions targeting inflammation may be an important approach to treat CVD combined with depression. However, the utility of inflammatory cytokines as biomarkers remains limited by their lack of disease specificity, given their elevation across multiple pathological conditions. Moreover, the therapeutic efficacy of anti-inflammatory interventions in CVD patients with comorbid depression awaits validation through large-scale randomized controlled trials.

## Endothelial dysfunction and platelet activation

Endothelial dysfunction and platelet activation play important roles in the relationship between CVD and depression. Under physiological conditions, endothelial cells regulate vascular tone, inhibit platelet activation, and maintain the integrity of the vascular wall by synthesizing and secreting various active substances such as nitric oxide (NO), endothelin-1, and prostaglandins (*Janaszak-Jasiecka et al., 2021*; *Pacinella, Ciaccio & Tuttolomondo, 2022*). However, clinical studies indicate that patients with depression exhibit higher plasma levels of platelet factor 4 (PF4) and beta-thromboglobulin (β-TG) after PCI compared to non-depressed patients (*Ning et al., 2024*), indicating that depression can promote platelet activation (*Reid et al., 2009*). The potential mechanism is that depression disrupts the homeostasis of the vascular endothelium through pathways such as activation of the SNS, leading to impaired physiological functions of endothelial cells and triggering endothelial dysfunction (*Lima et al., 2019*). When endothelial dysfunction occurs, the synthesis and release of key substances are reduced, weakening the inhibition of platelet activation and further promoting platelet activation. Activated platelets release large amounts of serotonin (5-HT), which promotes platelet aggregation and thrombus formation by binding to the 5-HT2A receptors on platelets. This process concurrently amplifies inflammatory factor release (*Williams et al., 2019*), ultimately accelerating AS, acute MI and other CVD (*Murphy et al., 2019*). It should be noted that there is currently a lack of longitudinal studies demonstrating whether improving endothelial function can simultaneously alleviate both depression and cardiovascular disease CVD.

## Brain-gut axis dysfunction

With the in-depth research on CVD combined with depression, the role of the brain-gut axis in the regulation of CVD and depression has also received attention (*Xu et al., 2024*). As shown in Fig. 3, the brain-gut axis refers to the bidirectional communication system formed between the brain and the gut through neural, endocrine, and immune mechanisms. Patients with depression often suffer from gut microbiota dysbiosis, which

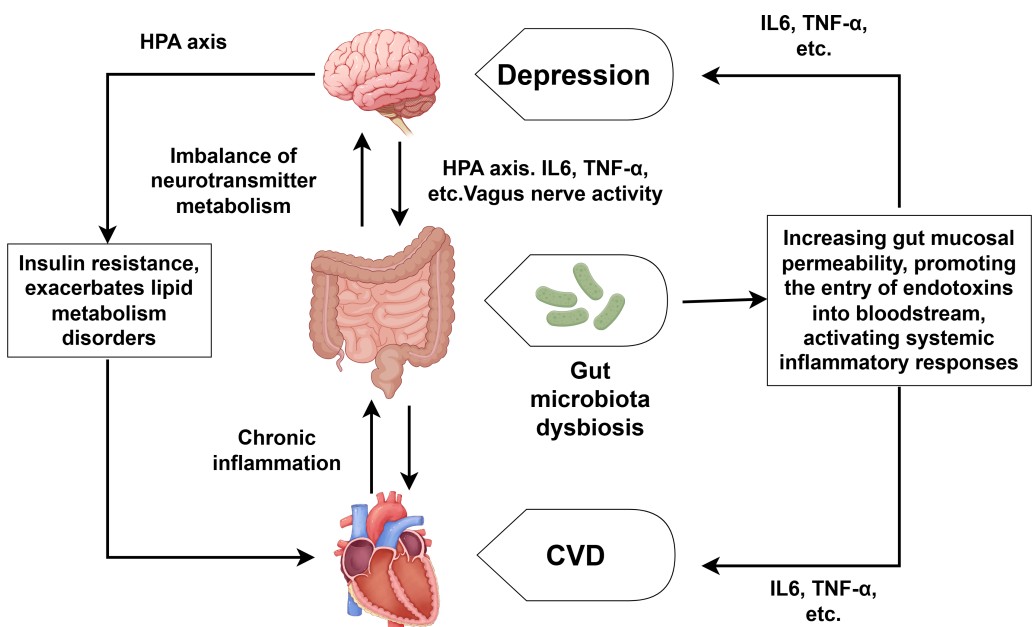

**Figure 3** **Comorbidity mechanisms between depression and CVD mediated through brain-gut axis dysfunction and lipid metabolism disorder.** Created with Figdraw 2.0: https://www.figdraw.com.

can lead to the imbalance of neurotransmitter metabolism (such as the synthesis of serotonin) and directly participate in the development of depression (*Liang et al., 2022*; *Rao et al., 2021*). Gut microbiota dysbiosis can also reduce short-chain fatty production, increase gut mucosal permeability, and facilitate endotoxin entry into the bloodstream. This process triggers systemic inflammatory responses, characterized by elevated levels of IL-6, TNF-α and other proinflammatory cytokine (*Filidou et al., 2024*; *Wei et al., 2024*). Moreover, the overactivation of the HPA axis can lead to increased CORT levels, which directly damage enterocytes and disrupt the integrity of the intestinal barrier. We already know that the overactivation of the HPA axis and elevated CORT can also lead to the substantial release of inflammatory factors such as TNF-α, IL-1 and so on (*Fioranelli et al., 2018*), which can promote the development of both CVD and depression. It is also worth noting that depression can inhibit the activity of the vagus nerve, affecting the transmission of signals along the brain-gut axis and further exacerbating gut microbiota dysbiosis (*Averina et al., 2024*; *You et al., 2024*). Conversely, CVD—especially its associated chronic inflammation—worsens intestinal microbiota disturbances, establishing a self-perpetuating pathological cycle. Notably, human intervention studies directly proving that specific gut microbiota alterations are the cause rather than the consequence of CVD-depression comorbidity remain scarce. Furthermore, the efficacy of probiotic/prebiotic interventions in improving cardiovascular and depressive symptoms among comorbid patients requires further validation through high-quality randomized controlled trials.

## Lipid metabolism disorder

Some scholars think that lipid metabolism disorders may represent a comorbid mechanism linking CVD and depression. Lipids, as essential components of cell membranes, are critical for maintaining normal cellular physiological functions. Depression disrupts cardiac lipid metabolism through multiple pathways, including neuroendocrine dysregulation and inflammatory responses. During depressive states, the overactivation of the HPA axis elevates cortisol CORT levels, which promotes lipolysis and insulin resistance, exacerbates lipid metabolism disorders, and subsequently causes myocardial energy supply imbalances, lipotoxicity damage, and electrophysiological disturbances (*Zhu et al., 2024*). Concurrently, depression-associated inflammatory responses mediate dysregulation of triglycerides, high-density lipoprotein and other lipids, thereby disrupting lipid homeostasis and exacerbating CVD—particularly AS (*Zhang & van der Vorst, 2024a*). Additionally, the impact of depression on the SNS also influence cardiac lipid metabolism. Since the pathological process of lipid metabolism disorders involves multidimensional interactive mechanisms, such as neuroendocrine dysregulation, immune-inflammatory activation, *etc.*, research on depression comorbid with CVD faces multiple complex challenges. It remains controversial whether lipid metabolism disorders are direct causal factors or merely downstream effects of CVD and depression. Thus, further evidence is required to substantiate their role as a core comorbidity mechanism.

## Other potential mechanisms

Unhealthy lifestyles and adverse work environments significantly increase the risk of CVD and precipitate the development of depression. Key factors include poor dietary habits, physical inactivity, smoking, excessive alcohol consumption, and chronic occupational stress and so on (*Yusuf et al., 2020*; *Baranova, Cao & Zhang, 2024*). Given that these lifestyle and environmental stressors ultimately mediate CVD and depression through core pathways—such as HPA axis dysregulation, systemic inflammation and other pathways— they should not be classified as comorbidities of CVD combined with depression.

Studies have reported that genetic susceptibility to depression correlates with heightened risks of CVD such as CHD and MI (*Li et al., 2022a*; *Lu et al., 2021*; *Zhang, Cao & Baranova, 2021*), suggesting shared genetic underpinnings between CVD and depression (*Amadio et al., 2024*). However, consider the possibility that genes may regulate depression, which in turn contributes to CVD pathogenesis. Therefore, whether genetic factors are the mechanism of comorbidity between CVD and depression remains contentious.

During the pathological process of CVD (*Zhang et al., 2024b*) and depression (*Zheng & Jin, 2024*), many microRNAs (miRNAs) show altered expression levels. Identifying miRNAs that can regulate the co-mechanism of CHD and depression may provide a new means for the treatment and diagnosis of CVD combined with depression. Unfortunately, research on miRNA-mediated comorbidity mechanisms remains nascent.

Despite the insufficient research on the mechanisms for simultaneously regulating CVD and depression, studies on the pathological mechanisms of either depression or CVD have been continuously progressing. This includes research on depression (*Bulluck et al., 2016*; *Del Re et al., 2019*) and research on CVD, particularly studies related to ferroptosis,

mitophagy, apoptosis, and so on (*Ding et al., 2020*; *Hursitoglu et al., 2023*; *Wang et al., 2023*). On the one hand, elucidating interactions between ferroptosis/mitophagy and established pathways (*e.g.*, HPA axis, autonomic nervous system) may hold significant importance for the treatment of CVD combined with depression. On the other hand, deep mechanistic investigation of either CVD or depression independently offers alternative biomarker discovery avenues for treating CVD combined with depression. For instance, reactive oxygen species (ROS) play a crucial role in the pathological processes of both CVD and depression, with NADPH oxidase (NOX) as a primary ROS source. Targeting NOX may thus confer dual therapeutic benefits effects in the treatment of CVD combined with depression (*Amadio et al., 2024*).

While the preceding sections detail discrete pathways, Table 1 consolidates key empirical support for the discussed mechanisms. As can be seen from the table, although clinical research on the impact of depression on CVD is abundant, literature investigating this relationship from a mechanistic perspective is scarce.

## CONCLUSIONS AND PERSPECTIVES

The comorbidities between CVD and depression are complex and interact with each other, involving many mechanisms such as autonomic nervous system imbalance, HPA axis abnormality, inflammatory response, brain-gut axis disorder and so on. Critically, these mechanisms operate within an interconnected network: HPA axis hyperactivity elevates glucocorticoids, driving pro-inflammatory cytokine release and endothelial dysfunction *via* reduced nitric oxide bioavailability; sympathetic nervous system overactivation amplifies HPA axis output while directly stimulating peripheral inflammation through catecholamine surges; and systemic inflammation concurrently disrupts lipid metabolism and gut-brain axis signaling through gut barrier impairment. This synergistic interplay creates a self-reinforcing vicious cycle that exacerbates both conditions. This self-reinforcing vicious cycle significantly exacerbates both conditions, posing substantial challenges to clinical diagnosis and management while profoundly impacting patient prognoses. Although mechanistic understanding has advanced, critical gaps persist—particularly in identifying specific biomarkers and comprehensively mapping comorbidity pathways. Future research should further deepen the understanding of comorbidity mechanisms, explore interdisciplinary cooperation models, and develop precise diagnostic tools. Through these efforts, it is expected to better treat CVD combined with depression, and contribute to the cause of global public health.

**Table 1  Evidence synthesis for comorbid mechanisms of depression and CVD.**

| Relevant diseases or mechanisms | Research type | Key findings | Reference |
|---|---|---|---|
| Depression and coronary heart disease | Review | Mental stress-induced myocardial ischemia is associated with activation in brain areas involved in the stress response and autonomic regulation of the cardiovascular system. | *Vaccarino et al. (2020)* |
| Depression and autonomic nervous system | Review | In a state of depression, sustained hyperactivity of the sympathetic nervous system leads to increased opening of calcium ($Ca^{2+}$) channels, which intensifies cardiac contraction load and triggers arrhythmias and ischemic events | *Del Rivero Morfin, Marx & Ben-Johny (2023)* |
| Depression and HRV | Clinical trial | Depressive symptoms were associated with lower HRV in HF patients, independent of physical fitness. | *Walter et al. (2019)* |
| Depression and HPA axis | Clinical trial | Stress impairs response to antidepressants via HPA axis and immune system activation | *Menke et al. (2021)* |
| Depression and HPA axis | Experimental reasearch | Suppressing the activity of CRH neurons in the paraventricular nucleus (PVN) can ameliorate myocardial ischemia-reperfusion injury (MIRI) | *Zhou et al. (2024)* |
| Depression and inflammatory response | Clinical trial | Metabolic and inflammatory factors might play a mediating role in the association between depression and premature CHD, especially central obesity. | *Lin et al. (2023)* |
| Depression and inflammatory response | Clinical trial | Depression is associated with increased C-reactive protein levels in patients with heart failure and hyperuricemia | *Wang et al. (2017)* |
| Depression and inflammatory response | Clinical trial | Inflammatory processes as well as attenuation of brain-derived neurotrophic factor (BDNF) availability are involved in the pathophysiology of major depressive disorder. IL-6 emerged as a robust positive predictor of BDNF only in the melancholic sample, wherein serum BDNF levels were accordingly enhanced. | *Patas et al. (2014)* |
| Endothelial dysfunction and platelet activation | Clinical trial | Experimentally induced mental stress induced platelet activation in patients with coronary artery disease. | *Reid et al. (2009)* |
| Depression with endothelial dysfunction and platelet activation | Clinical trial | Transient endothelial dysfunction with mental stress was associated with adverse cardiovascular outcomes in patients with coronary artery disease. | *Lima et al. (2019)* |
| Depression and Brain-gut axis dysfunction | Meta-analysis | By contrast, elevated L-glutamine degradation, and reduced L-glutamate and L-isoleucine biosynthesis were identified in depressive-associated microbiomes. | *Liang et al. (2022)* |
| Depression and Brain-gut axis dysfunction | Experimental reasearch | The anti-depression effects of fecal microbiota transplantation were associated with the suppressed activation of glial cells and NLRP3 inflammasome in the brain. | *Rao et al. (2021)* |
| Depression and lipid metabolism disorder | Experimental reasearch | Quercetin inhibited neuronal ferroptosis and promoted immune responses in BCRD mice by targeting the lipid metabolism-related gene PTGS2. | *Zhu et al. (2024)* |
| Depression and Genetic predisposition | Clinical trial | Genetic predisposition to depression may have positive causal roles on coronary artery disease and MI. | *Li et al. (2022a)* |

### Funding

This work was supported by the Public Hospital Reform and High-Quality Development of Traditional Chinese Medicine Project of Yichang City (YWGZ24-35, YWGZ24-03), Special fund for talents of Medicine and Health Sciences, Three Gorges University (2023JKXK05). The funders had no role in study design, data collection and analysis, decision to publish, or preparation of the manuscript.

### Grant Disclosures

The following grant information was disclosed by the authors:
The Public Hospital Reform and High-Quality Development of Traditional Chinese Medicine Project of Yichang City: YWGZ24-35, YWGZ24-03.
Special fund for talents of Medicine and Health Sciences, Three Gorges University: 2023JKXK05.

### Competing Interests

The authors declare there are no competing interests.

### Author Contributions

- Ling Huang conceived and designed the experiments, prepared figures and/or tables, and approved the final draft.
- Lei Zhang conceived and designed the experiments, prepared figures and/or tables, and approved the final draft.
- Cuihua Liu performed the experiments, analyzed the data, prepared figures and/or tables, and approved the final draft.
- Qiang Xu performed the experiments, authored or reviewed drafts of the article, and approved the final draft.
- Kuncheng Qiu analyzed the data, authored or reviewed drafts of the article, and approved the final draft.

### Data Availability

This is a literature review.

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
