# Peer review of "Interaction and mechanisms of depression and cardiovascular disease: a mini-review"

_PeerJ, doi:10.7717/peerj.20148_

## Round 0.1 · original submission · Minor Revisions

· Academic Editor

Minor Revisions

Dear Authors,

Thank you for submitting your manuscript to PeerJ.

The reviewers made some suggestions to improve the quality of the article. Please revise the manuscript taking into account all these comments and suggestions.

Kind regards,
Marialaura Di Tella

**Language Note:** The review process has identified that the English language must be improved. PeerJ can provide language editing services - please contact us at [email protected] for pricing (be sure to provide your manuscript number and title). Alternatively, you should make your own arrangements to improve the language quality and provide details in your response letter. – PeerJ Staff

·

Basic reporting

The Authors made a systematic PubMed search for their purposes. The manuscript is well written and structured. The topic is fascinating. The figures are appealing. A table, or more, with study summaries is advisable.

Experimental design

The idea of carrynig out a systematic search is advisable, but the authors should make clear that they underwent a systematic revision of the literature. If they do not, they should explain why in the introduction and what the actual scope of this review is. In particular, the audience they are expecting to reach.

Validity of the findings

The findings and the area explored are comprehensive of the topic and subtopics.

Additional comments

English grammar revision and a double check for typos are required.

·

Basic reporting

I would like to thank you for the opportunity to review your manuscript on the interaction and mechanisms of depression and cardiovascular disease (CVD). The topic you have explored is undeniably important, particularly in light of the growing recognition of the bidirectional relationship between these two conditions and their significant impact on global public health.

While this manuscript presents an important subject, I believe that further refinement of these aspects would be necessary to ensure a more substantial and impactful contribution to the field.

Experimental design

The article explores several biological mechanisms, but it tends to focus on theoretical pathways with limited empirical support. A more critical evaluation of the literature would strengthen the manuscript's contribution to our understanding of the complex interactions between depression and CVD. The treatment strategies, while briefly mentioned, are not explored in sufficient detail to provide clinicians with actionable insights. Given the significant clinical challenges posed by this comorbidity, a more thorough discussion of current treatments and research gaps would be valuable.

Validity of the findings

The author may consider a clearer explanation of the interconnections between the different pathophysiological mechanisms, as well as a more focused discussion on patient outcomes. The lack of this integration weakens the clinical relevance of the review, as it does not fully address the impact of these conditions on patient prognosis and quality of life.

Additional comments

Addressing these limitations should improve this review. I wish you continued success in this work. I recommend "minor revision" for this manuscript.

·

Basic reporting

The authors reviewed the potential mechanisms behind the comorbidity between depression and CVD. These findings may have the potential to facilitate the treatment of CVD combined with depression. I have some recommendations for the authors:

1. The authors need to discuss causal relationships between depression and CVD, as demonstrated previously (PMID: 34859065). Drug use is a vital confounding factor that needs to be discussed, as a previous study showed that it is antidepressants that boost the risk of CVD (PMID: 38490691).

2. Some confounding factors can be discussed, such as education and income, given that they are vital risk factors for CVD and mental disorders (PMID: 38440407).

Experimental design

The study design is appropriate.

Validity of the findings

The findings are solid.

---

## Round 0.2 · accepted · Accept

· Academic Editor

Accept

Dear Authors,

Thank you for submitting your manuscript to PeerJ.

The Reviewers have positively evaluated the paper; therefore, it can be accepted for publication.

Kind regards,
Marialaura Di Tella

·

Basic reporting

I have no further questions.

Experimental design

I have no further questions.

Validity of the findings

I have no further questions.

Additional comments

I have no further questions.

·

Basic reporting

The authors have addressed my comments in a satisfactory manner.

Experimental design

Study design is sound.

Validity of the findings

no comment